# Phenotype Alterations in the Cecal Ecosystem Involved in the Asymptomatic Intestinal Persistence of Paratyphoid *Salmonella* in Chickens

**DOI:** 10.3390/ani13182824

**Published:** 2023-09-06

**Authors:** Michael H. Kogut, Mariano Enrique Fernandez Miyakawa

**Affiliations:** 1Southern Plains Agricultural Research Center, USDA-ARS, College Station, TX 77845, USA; 2Instituto de Patobiología, Instituto Nacional de Tecnología, Nicolas Repetto y Los Reseros S/N, Hurlingham 1686, Buenos Aires, Argentina; fernandezmiyakawa.m@inta.gob.ar

**Keywords:** microbiota, gut health, disease tolerance, T regulatory cells, enteric nervous system

## Abstract

**Simple Summary:**

To combat infections, hosts employ a combination of antagonistic and cooperative host defense strategies. The former refers to pathogen killing mediated by host immunity (disease resistance), while the latter refers to physiological defense mechanisms that promote host health during infection independent of pathogen killing, leading to a cooperation between the host and the pathogen (diseases tolerance). In chickens, the paratyphoid S*almonella* evolved the capacity to survive the initial robust immune response and persist in the avian ceca for months without triggering clinical signs. The persistent or carrier phase of a *Salmonella* infection in the avian host involves a complex balance of antagonistic and cooperative host defense strategies. Initially, the host reacts with a normal inflammatory response that controls bacterial invasion. After 3–4 days of inflammation, the host response changes to a more anti-inflammatory response characterized by changes in the local intestinal physiology that is no longer antagonistic to the bacterial pathogen, but instead ‘tolerates’ its presence. Thus, the chicken becomes a carrier of the pathogen allowing it to survive in the intestine without causing disease. It is hoped that understanding these mechanisms of pathogen survival in the chicken will allow future research to exploit these novel disease tolerance mechanisms to impact poultry health and reduced foodborne infections of *Salmonella*.

**Abstract:**

The gastrointestinal ecosystem involves interactions between the host, gut microbiota, and external environment. To colonize the gut of poultry, *Salmonella* must surmount barriers levied by the intestine including mucosal innate immune responses and microbiota-mediated niche restrictions. Accordingly, comprehending *Salmonella* intestinal colonization in poultry requires an understanding of how the pathogen interacts with the intestinal ecosystem. In chickens, the paratyphoid S*almonella* have evolved the capacity to survive the initial immune response and persist in the avian ceca for months without triggering clinical signs. The persistence of a *Salmonella* infection in the avian host involves both host defenses and tolerogenic defense strategies. The initial phase of the *Salmonella*–gut ecosystem interaction is characteristically an innate pro-inflammatory response that controls bacterial invasion. The second phase is initiated by an expansion of the T regulatory cell population in the cecum of *Salmonella*-infected chickens accompanied by well-defined shifts in the enteric neuro-immunometabolic pathways that changes the local phenotype from pro-inflammatory to an anti-inflammatory environment. Thus, paratyphoid *Salmonella* in chickens have evolved a unique survival strategy that minimizes the inflammatory response (disease resistance) during the initial infection and then induces an immunometabolic reprogramming in the cecum that alters the host defense to disease tolerance that provides an environment conducive to drive asymptomatic carriage of the bacterial pathogen.

## 1. Introduction

A feature of paratyphoid *Salmonella* serovars (broad-host range) in poultry is a persistent intestinal infection, or asymptomatic carrier state [1,2,3,4]. Further, these *Salmonella* asymptomatic carriers excrete high numbers of *Salmonella* into the environment [1,2,3,4], thereby increasing their propagation and facilitating contamination of other birds by horizontal transmission or affecting newly hatched chicks [5,6].

The establishment of persistence in the chicken gut by *Salmonella* requires overcoming both host-mediated elements such as a substantial immunometabolic mucosal response [7], but also microbiota-linked features such as production of antimicrobial fermentation products and the occupation of nutrient and adhesion niches [8,9]. Surmounting both features of the avian intestinal ecosystem by *Salmonella* is quite different than that observed in the human and swine gut [9,10,11]. The mechanisms that underlie pathogen persistence in the gut form the basis for this review.

## 2. The Gut Ecosystem

Optimal gut health is of vital importance to the performance of production animals, including poultry, to be able to perform to their genetic potential. Understanding a ‘healthy gut’ requires knowledge of the functional interactions of all components of the enteric ecosystem: the host, the microbiota, and the environment. The connections between these diverse physiological features of the enteric ecosystem underscore the extent of areas encompassed by the gut and the difficulty in correlating specific components of gut health with the ability to regulate poultry performance.

The gut can be considered as a complex and dynamic ecosystem molded during the interactions between the host, the gut microbiota, and environmental factors, including diet, temperature and humidity, animal density, infections, and mycotoxins. A diverse number of bacteria, commensal, potential beneficial, and pathogenic, must overcome several physical, chemical, and biological barriers imposed by this ecosystem, including host immune responses, the epithelial cell barrier, and microbiota-mediated events [12]. Consequently, bacterial intestinal colonization requires understanding the mechanisms by which the gut ecosystem interacts with microbes attempting to associate with the established community.

### 2.1. Components of the Intestinal Ecosystem

In general, there are three foundational components of the intestinal ecosystem: (a) host factors, (b) microbiota-linked factors, and (c) environmental factors, as recently detailed by Barron and Young [13].

#### 2.1.1. Host Factors

The host provides the physical and biochemical foundations of the intestinal ecosystem with the intestinal epithelium and mucosal immune system forming the basis of the gut milieu.

(A)Intestinal epithelium

The epithelium physical firewall is a single layer of epithelial cells that separate the densely colonized, and environmentally exposed, intestinal lumen from the largely sterile subepithelial tissue. The intestinal epithelial cell layer displays a number of distinctive functions including production of antimicrobial peptides and the secretion of mucus secrete antimicrobial peptides (defensins, cathelicidins, C-type lectins), which are a key defense against luminal microbes. Linking the epithelial cells are tight junctions, which help form a continuous luminal surface and help seal the intercellular space, near the apical surface, from the external environment [14]. Besides being the primary barrier preventing a microbial breach of the intestine, the epithelial cells should also be considered part of the cellular component of the innate immune response possessing pattern recognition receptors (PRRs) for sensing microbial-associated molecular patterns (MAMPs), but also capable of producing cytokines and chemokines to drive an inflammatory response against pathogen infection.

(B)Immune system

Below the epithelial layer is the final component of the intestinal barrier: the immunological barrier where the professional immune cells (macrophages, DC, and lymphocytes) reside in the lamina propria [14]. This intestinal immune barrier has two distinct functions: the ability to respond to pathobionts (potential pathogenic microbes), invasive pathogens, and microbial products while also maintaining a state of tolerance to the diverse and beneficial commensal intestinal microbes [15]. Both systems working together through innate immune sensing using PRRs on epithelial cells and professional immune cells in the lamina propria (dendritic cells (DC) and macrophages) trigger immune pathways resulting in microbial killing and the activation of various acquired immune effector T cells (Th1, Th2, Th17, and Treg) all while keeping the resident microbiota in check without generating an overt inflammatory response. IgA-producing plasma cells, intraepithelial lymphocytes, and γδT cell receptor-expressing T cells are lymphocytes that are uniquely present in the mucosa. In addition, of the γδT cells in the intestinal lamina propria, there are significant numbers of IL-17-producing T cells and regulatory T cells (reviewed in [14]).

(C)Enteric neuroendocrine system

The gut is the largest neuroendocrine organ in the body owing to the large numbers of neurons, gut hormones, and secondary messengers involved in regulating physiological functions in the host [16,17]. The neuroendocrine system (NES) of the gut involves two components including the gut endocrine cells, which are in the gut mucosa, and the enteric nervous system (ENS) in the gut submucosa. This system regulates several functions of the GI tract, such as motility, secretion, absorption, microcirculation in the gut, local immune defense, and cell proliferation [16,17]. The ENS comprises a large variety of neurotransmitters and associated receptors.

The gut contains a large number of enterochromaffin cells (endocrine cells that produce serotonin) dispersed among the epithelial cells of the gut mucosa in the intestine of the chicken [18,19]. The gut endocrine cells secrete signaling peptides into the lamina propria of the gut lining, where they have regulatory activity on the enteric nervous system (ENS), afferent and efferent nerve fibers of the central nervous system (CNS), and the autonomic nervous system (reviewed in [19]). Further, neurochemicals play a recognized role in determining bacterial colonization and interaction with the gut epithelium [20].

The gut–brain axis is a bidirectional information exchange network that connects the gut, the enteric endocrine system, and the CNS to the brain [21]. Villageliu and Lyte [22] corroborated the presence of the gut–brain axis in chickens, albeit not functional characterized. However, both Wickramasuriya and colleagues [23] and Cao et al. [24] have described the effects of environmental stresses on the gut–brain axis functional regulation of the physiology of the chicken.

#### 2.1.2. Microbiota Factors

The chicken gastrointestinal (GIT) tract is home to a complex microbial community that links the environment to the health status of the host. The avian commensal microbiota are strategic managers of host physiology involved in regulating bird health [25,26], directing host intestinal metabolism and immunity, and directing a metabolome that affects energy balance and body weight [27]. Lastly, the residential microbes in the gut play a significant role in inhibiting pathogens from colonizing by a process called colonization resistance [8,9].

(A)Microbial composition: nutrient competition

Effective nutrient acquisition in the competitive environment of the gut is essential for persistence of indigenous microbes. Nutrient resources for the microbiota are provided from dietary components or metabolites produced by the host either from the diet or mucus secretions [28,29]. Indigenous microbiota utilize dietary amino acids, carbohydrates, essential trace metals (iron, zinc, copper, manganese), and respiratory electron acceptors (O_2_ and NO_3_^−^), thereby starving the pathogens of essential nutrients and molecules [29,30,31].

(B)Microbial composition: site competition

Commensal bacteria are able to control host membrane glycosylation and/or use it as a nutrient, thus creating a novel niche that reduces pathogen access to the epithelial barrier [32]. Further, commensal microbes occupy potential binding sites on the intestinal epithelium by deploying numerous molecular structures including outer membrane proteins, capsules, lectins, adhesins, and fimbriae [33]. Interestingly, Donaldson and colleagues [34] have shown that some symbiotic bacteria can co-opt the secretory IgA response to mediate stable colonization of the intestine which excludes colonization of indigenous pathobionts and pathogens access to the intestine.

(C)Microbial composition and antimicrobial peptides

Commensal microbes can limit enteric bacterial colonization through direct microbe-microbe interactions that include contact-dependent killing competitor bacteria via the type 6 secretion system (T6SS), suppression of competitor bacteria growth by contact-dependent secretion of effector proteins which bind to specific receptors on the competitor bacteria that activate a toxic effector domain, or the production of bacteriocins which form pores in the competitor symbionts, pathobionts, or pathogens which induces leakage of cellular contents [32,35].

(D)Microbial metabolite production

The microbiota, using a number of biochemical pathways, metabolize diet- and host-derived metabolites that can have a direct impact on the intestinal immune system and inhibit colonization of the intestine by competitor bacteria. For example, bacterial metabolites such as short chain fatty acids (SCFA) serve as an energy source to the epithelial cells but also have antimicrobial activity and limit virulence factor expression on pathogenic bacteria [36,37,38]. Further, microbiota can degrade dietary tryptophan to promote epithelial cell barrier function and breakdown dietary arginine which inhibits pro-inflammatory cytokine production [39].

## 3. *Salmonella* Interactions with the Intestinal Ecosystem in Chickens

One of the major causes of human gastroenteritis is *Salmonella enterica* (*S. enterica*) due to infected poultry products. Serovars, such as *S*. Typhimurium and *S*. Enteritidis, cause acute gastroenteritis in humans, but these bacteria colonize the intestines of chickens without causing disease [40]. This subclinical colonization poses a challenge for the prevention of foodborne transmission of *Salmonella* since colonized poultry are difficult to identify. The basis of the difference in the clinical consequences of *S*. Typhimurium and *S*. Enteritidis infection between mammals and birds is still vague, but recent studies have begun to unravel the fundamental differences which include the differential expression of virulence genes [41,42,43], host inflammation [42,44,45], and bacterial dietary and metabolic requirements [10,46,47].

### 3.1. Salmonella and Host Factors

#### 3.1.1. Immune System

Paratyphoid Salmonella have evolved a unique survival strategy in poultry by surviving the initial robust immune response and persistently infecting the intestine. This chronic colonization of the intestinal tract is an important aspect of persistent Salmonella infection because it results in a silent propagation of bacteria in poultry stocks due to the impossibility to isolate contaminated animals. Data from our lab promote the hypothesis that Salmonella have evolved a unique survival strategy in poultry that minimizes host defenses (disease resistance) during the initial infection and then exploits and/or induces a dramatic immunometabolic reprogramming in the cecum that alters the host defense to disease tolerance.

(A)Disease Resistance

Disease resistance is the host defense strategy grounded on the immune response’s capacity to detect and eliminate pathogens, i.e., host immunity [48,49,50]. The initial phase of the chicken cecum to *Salmonella* infection is manifested by the absence of clinical disease and functions to control pathogen invasion and reduce or eliminate the invading pathogen [48]. Specifically, *Salmonella* infection: (1) activated both Toll-like receptor (TLR) and Nod-like receptor (NLR) signaling pathways to initiate an innate immune response; (2) induced the production of chemokines CXCLi2 (IL-8) and cytokines IL-2, IL-6, IFN-α, and IFN-γ; (3) induced the phosphorylation of Janus kinase (JAK)/signal transducer and activator of transcription (STAT) signaling pathway that initiates innate immunity and coordinates adaptive immune mechanisms; (4) triggered both the intrinsic and extrinsic apoptotic pathways; and (5) activated the T cell receptor signaling pathway through the AP-1 and NF-κB transcription factor cascades [48].

(B)Disease Tolerance

Disease tolerance is a second host defense strategy that limits the damage caused by a pathogen’s growth without affecting or reducing pathogen numbers [51,52]. In chickens, a second phase (starting at 4 days post-infection) of *Salmonella* persistence is characterized by an increase in the CD4+ CD25+ T cell (T regulatory [Tregs]) population in the cecum of *Salmonella*-infected chickens. Functionally, the cecal Tregs had increased suppressive activity for T effector cells and had a profound increase in IL-10 mRNA transcription [49,50].

Using chicken-specific kinomic immune-metabolism peptide arrays and quantitative real-time PCR of *Salmonella*-infected cecal tissue 4 to 17 days post challenge, distinct immune and metabolic pathways are altered and changed the local immunometabolic environment. In general, two energy sensory kinases, AMPK and mTOR, are key players linking specific extracellular milieu and intracellular metabolism. Phenotypically, the early response (4 to 72 h) to *Salmonella* infection is pro-inflammatory, fueled by glycolysis and mTOR-mediated protein synthesis, whereas after 4 days post-infection, the local environment has undergone an immune-metabolic reprogramming to an anti-inflammatory state driven by AMPK-directed oxidative phosphorylation [51,52].

#### 3.1.2. Paratyphoid *Salmonella* Manipulation of the Enteric Nervous System

(A)*Salmonella* and neurochemical release

The Mellata lab at Iowa State University conducted a series of experiments to understand interactions between the nervous and immune systems during paratyphoid *Salmonella* infections in chickens [53]. Using the drug reserpine, which induces the release of intracellular storage of catecholamines like norepinephrine (NE) to treat cecal explants and isolated Tregs from chickens, NE was increased [53]. Further, reserpine treatment in vivo reduced the level of intestinal *Salmonella* Typhimurium and other Enterobacteriaceae. These results provided the first direct data that *Salmonella* colonization inhibits the release of neurochemicals that participate in the regulation of the enteric neuro-immunological responses to infection [53].

(B)*Salmonella* and neurotrophin signaling

Neurotrophins are a family of growth factors critical to the functioning of the nervous system including neuron formation and survival. Neurotrophies are ligands of Trk receptors. Trk receptors are a family of tyrosine kinases that regulate gut sensation, motility, and epithelial barrier function, and protect enteric neurons and glial cells from damaging insults in the microenvironment of the gut via several signaling cascades including the extracellular signal-regulated kinases (ERK) pathway, the phosphoinositide 3-kinase (PI3K)/Akt pathway, and the phospholipase C (PLC)-γ pathway. *Salmonella* infection of the chicken cecum dephosphorylates the Trk A and C receptors resulting in the dephosphorylation of the ERK pathway, PI3K/Akt pathway, and the PLC-γ pathways (Figure 1). These results provide evidence that *Salmonella* infection manipulates enteric neuron functionality during colonization of the cecum, thus blocking the gut–brain axis that controls the enteric host response to the pathogen.

(C)Infection-induced feeding behavior

A common feature of enteric infections is a reduction in feeding behavior or anorexia. However, paratyphoid *Salmonella* colonize the intestines of chickens without causing disease [40] including a lack of effect on growth feed intake, growth, and production [54]. As a function of disease tolerance host response, it has been shown that *Salmonella enterica* manipulates the gut–brain axis to inhibit anorexia, which reduces its virulence but promotes its transmission [52,54,55].

#### 3.1.3. Salmonella and Microbiota-Linked Factors

(A)Gastrointestinal microbiota

The crop, proventriculus and gizzard, duodenum, jejunum, and ileum (small intestine), ceca, large intestine, and cloaca is the basic structure of the poultry gastrointestinal tract [31] and each part plays a different role that is influenced by the dynamics of the microbiota. The gut chicken microbiota at various stages of the animal’s life supports functions that range from protection against pathogens and nutrient production to maturation of the immune system. *Salmonella* face an important competition for space and nutrients with this large and diverse community of gastrointestinal microorganisms.

Ileal and cecal microbiomes have been characterized in commercial chickens as a means of evaluating the influence of the microbiota on performance, health, and disease [25,27,56,57]. Microbiota composition is affected by numerous factors such as sex [58], genetics [59], diet [60], environment [25], and host stress [61,62]; yet, age is consistently the primary driving factor with the microbiota undergoing successional changes during the life cycle of a bird [63,64]. The richness and the composition of microbiota strongly influence the gut ecosystem functioning. As the individual grows, the gut microbiota undergo dynamic changes with greater diversity and complexity as the host ages, and variably by intestinal tract segments [56]. To describe and investigate the dynamics of the microbiota or to detect changes in composition, usually mathematical tools such as α-diversity indices (diversity within a sample) and β-diversity indices (diversity among samples) as well as the description of taxonomic composition are used. *Salmonella* colonization impact both diversity and composition of chicken gut microbiota [65,66]. However, it must be considered that the colonization and the interaction of *Salmonella* with the microbiota are complex processes that depend not only on the characteristics of the pathogen but also on other variables such as the time of infection, host characteristics, nutrition, and the gut microbiota composition itself.

(B)Microbiota and *Salmonella* intestinal colonization

Considering microbiota composition and timing, the intestine of newborn chicks is a relatively sterile environment and an excellent opportunity for certain pathogens (such as *Salmonella*) to rapidly colonize and spread freely in the intestine [67]. Proteobacteria, primarily Enterobacteriaceae, including members such as *Escherichia*, *Shigella*, *Enterococcus*, and *Salmonella*, are the most common genera detected in day-of-hatch birds, but also high abundance of Enterobacteriaceae has been observed in and at 3-day-old layer chickens [65,68]. Proteobacteria abundance decreased significantly with age as Firmicutes members, particularly Clostridia, and overall diversity increase [69]. Firmicutes increase in abundance and taxonomic diversity starting around day 7 [63]. The reduction of certain bacteria taxa from Enterobacteriaceae family during the early post-hatch period might enhance the host resistance to enteric pathogen infection as well as transient perturbation of the gut microbiota produced by different stressors including antibiotics [70]. Although *Salmonella* is a common gut colonizer in poultry [71], it had become evident that the taxonomic features of the microbiota are an important factor determining susceptibility and resistance to *Salmonella* colonization at the individual level [72].

*Salmonella* enters chickens through vertical transmission from infected hens or via the oral route through infected feed, water, or litter, and colonize the distal part of the ileum and cecum [73]. Oral inoculation of *Salmonella* after a couple of days after hatch resulted in an important shift in chicken cecal microbiota composition at 7 and 14 dpi [74]. Experimentally, *Salmonella* is able to induce an asymptomatic carrier state when 5 × 10^4^ colony-forming units (CFU) are orally inoculated in chicks [75], but the same bacteria load in older chicks has no effect and it is required up to 1 × 10^8^ CFU to reproduce a successful infection in 30-week-old hens [76], supporting the observation that hens are usually more resistance to *Salmonella* infection than chicks. These observations suggest that *Salmonella* colonization in the gastrointestinal tract of the chickens has a direct effect on altering the natural development of the gut microbiota.

The role of the microbiota is also important to explain the heterogeneity of infection associated with the presence of super-shedders which constantly disseminated *Salmonella* to the low-shedder chicks [77]. The shedding levels are highly influenced by gut microbiota composition at the moment of infection, with α-diversity indices correlating with the shedding level as low-shedder chicks showed the lowest α-diversities [72]. The homogeneity of microbial compositions within the shedding level was corroborated by the analysis of β-diversity indices. Differences in the composition of the intestinal microbiota show an important influence on the susceptibility or resistance to colonization by *Salmonella* as adult hen microbiota samples orally delivered may have a protective effect on one-day-old chicks but the absence of gut microbiota results in super-shedders animals after *Salmonella* colonization [72,78]. The outcome of the first exposure to *Salmonella* seems to be determinant for shedding as, contrary to super-shedder chicks, low shedders can block early colonization [77].

#### 3.1.4. *Salmonella* Enterica Serovar Enteritidis (SE) and Microbiota

In the immediate post-hatch period in chickens, SE infection is disadvantageous to the expansion of the gut microbiota resulting in the reduction in microbial diversity and an increase in potential pathogens in the microbial community [65,79]. In contrast, SE infection in older animals did not show dominance of a specific species in the community when compared to non-infected animals as determined by Shannon index (α-diversity) [79]. However, 3 weeks after infection of SE in these older hens, an increased colonization by minority members of the community following the infection as observed in the changes of Chao1 index (α-diversity) was observed [79]. The impact of SE in microbial communities in challenged one-day-old layer chicks seems to be more substantial in later stages of the infection [65]. However, besides that SE can be considered a good gut competitor and bacterial colonization was exclusively localized in the cecum of the infected chicks, systemic infection was not observed in an oral challenge model, also when high doses of inoculum were administered [79].

SE infection in young chicks significantly reduces the overall diversity of the microbiota population due to the expansion of the *Enterobacteriaceae* family; however, infection had a more significant impact on microbial communities during the later stages of infection where a negative correlation between *Enterobacteriaceae* and *Lachnospiraceae* [70], *Ruminococcaceae*, *Erysipelotrichaceae*, and *Peptostreptococcaceae* was observed in SE challenged chicks [65]. Although SE infection in newly hatched chickens did not influence the predominant cecal microbiota, an increase of *Lactobacillaceae* was observed [70]. The *Ruminococcaceae* family, which is more abundant in non-infected animals, has been suggested as a signature of *Salmonella* infection [5,80]. Considering only the genera presenting a high relative abundance (>5%), *Enterococcus* genus was considered the main taxonomic feature allowing to predict the low- or super-shedder phenotypes, although SE colonization was not modified by oral inoculation of chicken strains of *E. faecium* [72]. Although *Enterococcus* spp. might not prevent *Salmonella* colonization by itself, their proportional increase can be associated with another change at microbial or host physiologic level and could be used as biomarker of *Salmonella* infection. However, although not opposed to the previous observation, the association of *Enterococcus* with well-defined commensal bacteria (*E. coli*, *Clostridium*, *Lactobacillus*) orally inoculated the day of hatch reduced SE excretion and increased the proportion of low-shedders animals, which was not observed when inoculated separately [72].

#### 3.1.5. *Salmonella* Enterica Serovar Typhimurium (ST) and Microbiota

ST colonization of the chicken intestinal tract, either after experimental challenge or natural infection, alter microbiota composition with resultant decreases in cecal *Enterococcus*, *Lactobacillus*, *Escherichia*, and Bacillaceae [81]. Further, ST infection significantly reduced α-diversity indexes of ileal microbiota of broiler chickens [54] with a notable increase in *Escherichia-Shigella* genus levels [82].

A few days after ST colonization (3 days post-infection), an increased α-diversity was observed in cecal microbiota, but the change was rapidly inverted at 5 and 7 dpi, suggesting a differential effect depending on the dynamics of the innate immune response [83]. Additionally, the challenge decreased the cecal abundance of *Lactobacillus*, *Bifidobacterium*, *Trabulsiella*, *Oscillospira*, *Holdemania*, *and Coprococcus*, and *increased Klebsiella* and *Escherichia* [66,83]. In the feces of ST-challenged laying chickens, a significantly lower microbial α-diversity was observed when compared to control during several weeks after the challenge with a decrease in the abundance of *Blautia*, *Enorma*, *Faecalibacterium*, *Shuttleworthia*, *Sellimonas*, *Intestinimonas*, and *Subdoligranulum* and increase in the abundance of *Butyricicoccus*, *Erysipelatoclostridium*, *Oscillibacter*, and *Flavonifractor* [83].

(C)Intestinal microbial functions and host metabolite production

Alteration in the overall microbial community following *Salmonella* infection in chickens has a consequent effect on the host regulation of cecum-associated metabolic networks [65,84,85,86]. Metagenome functional prediction of the chicken gut microbiome shown various altered pathways in 2-week-old SE-infected animals, including functional genes associated with ribosomal activity and nucleotide metabolism (purine and pyrimidine) which could suggest an interference of *Salmonella* with the metabolism of intestinal microorganisms and intestinal activity [79]. Metabolic pathway analysis of the cecal content from birds infected with SE revealed a disruption in microbiota metabolic pathways related to arginine and proline metabolism as well as reduced tricarboxylic acid cycle (TCA) activity. Similarly, ST infection of chick early post-hatch also found revealed differences with non-infected animals in metabolic composition of ceca content including lactate, the main product of glucose fermentation of *Enterococcus* [87], supporting the observation that enterococci are significant members of the cecal microbiota during *Salmonella* infection. Furthermore, the microbiota composition was unchanged in neonatal chicks infected with ST, but the functional activity of the microbiota was dramatically altered [86]. For example, ST infection induced the increase expression of genes involved in branched-chain amino acid (BCAA) production, such as leucine, isoleucine, and valine [86] which play a key role in the growth, production performance, immunity, and intestinal health of chickens [87,88].

Global gene expression pattern and metabolites profile in the host is altered during *Salmonella* infection. A significant accumulation of metabolites was identified in the ileum and cecum of *Salmonella*-infected birds [79,86]. Immediately after infection, a comparatively moderate number [6] of metabolites were affected in cecum of infected chickens. However, a week after infection, a considerable number of metabolites [78] are altered but the difference was dramatically reduced (three metabolites) a week later [79]. Up-regulation of arginine and proline metabolism was detected in association with *Salmonella* infection, suggesting the activation of a host metabolic adjustment strategy to reduce the intestinal inflammation during *Salmonella* infection to improve their intestinal colonization [79]. Arginine is a common amino acid substrate used by the inducible nitric oxide synthase (iNOS) for nitric oxide production, one of the key innate immune responses to induce inflammation as part of the host defense mechanism [89]. The analysis of the metabolites revealed reduced tricarboxylic acid cycle (TCA) in the *Salmonella*-infected group compared to the non-infected group, suggesting that a change in host cellular energy metabolism during *Salmonella* infection occurred [79]. The alteration in host metabolic response could be associated with the innate immune activation triggered by LPS stimuli from *Salmonella* infection [90].

## 4. Concluding Remarks

Because of the increase in antimicrobial-resistant microbes, the use of antibiotics as growth promoters has either been banned by government intervention or removed by producers due to the consumer demand for ‘no antibiotics ever’ or ‘raised without antibiotics’ poultry products. Thus, there is an ongoing demand for the development and use of alternatives to antibiotics for growth promotion and food safety.

Herein, we have provided a review of the literature on *Salmonella* interactions with the different components of the avian GIT ecosystem. Further, we have shared insights into how these interactions appear to be involved in the establishment of a persistent *Salmonella* infection in the cecal lumen. The dynamics of these host–pathogen interactions involve host neuroimmune and immunometabolic pathways, pathogen virulence, gut microbiota, and the crosstalk between all components which has been described by Troha and Ayres [91] as ‘a household of three’. In this context, Troha and Ayres [91] described that the metabolic needs of all three members of the household must be studied to develop novel strategies that target metabolic processes that underpin the intestinal ecosystem.

Traditionally, studies into the host response of poultry to infections with paratyphoid *Salmonella* serovars have concentrated on host resistance mechanisms which target the elimination of the pathogen by the immune system [46,48,51,52]. However, in poultry, paratyphoid *Salmonella* have evolved the capacity to survive this initial immune response and persist in the gut lumen for weeks without causing clinical disease in birds [40,48]. This persistent colonization of the intestinal tract is an important aspect of a *Salmonella* infection because it results in the silent propagation of bacteria in poultry stocks due to the impossibility to isolate contaminated animals. This persistence suggests that a second defense mechanism has evolved in chicken-*Salmonella* infection biology that functions to foster host health instead of removal of the pathogen [51,52]. This alternate defense strategy, known as disease tolerance, involves protecting the overall physiological homeostasis in the bird [51,52,53]. Collectively, the information from the literature implies that these neuro-immunometabolic connections between the host and its microbiota could be manipulated and that targeting the regulators of these neuro-immunometabolic pathways signify a promising translational approach to novel therapeutics in the future.

The use of omics technologies has opened the doors for better understanding of the *Salmonella*–intestinal ecosystem interactome, but more holistic approaches are required. All-inclusive studies using systems biology approaches are needed. For example, we have cited a number of studies describing the neuroimmune, immunometabolic, and microbiological alterations induced in the cecum of the host by *Salmonella* infections but provided no definitive information on the role that the microbiota play in this environment. Likewise, dozens of studies have provided ‘lists’ of microbiota compositional changes that have been found during *Salmonella* infections, but few, if any, of these studies showed a causal relationship to the host functional responsiveness to the infection. Our hope is that providing this overview in the literature on the *Salmonella*–intestinal ecosystem interactome will encourage more collaborative studies between laboratories concentrating on these interactions between the avian gut, the gut microbiota, and *Salmonella.*

## Figures and Tables

**Figure 1 animals-13-02824-f001:**
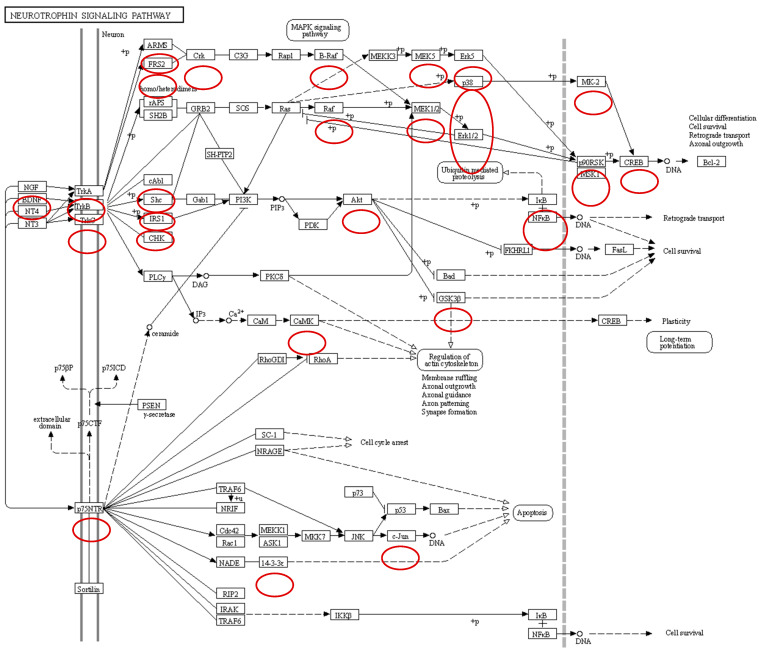
Effect of *Salmonella* infection on neurotrophin signaling pathway in cecum. All peptides encircled are significantly dephosphorylated during *Salmonella* infection when compared to non-infected cecal tissue.

## Data Availability

Not applicable.

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
