# Peer review of "Phenotype Alterations in the Cecal Ecosystem Involved in the Asymptomatic Intestinal Persistence of Paratyphoid Salmonella in Chickens"

_animals, 2023, doi:10.3390/ani13182824_

Round 1

Reviewer 1 Report

Manuscript Draft ID:  animals-2550598-peer-review-v1

REVIEW:

 Summary: The analysis of interactions between poultry chicken immune system, their gut microbiota and environmental conditions over the colonization and persistence of Salmonella in Poultry constitute an important item to review. Salmonellosis is one of the most critical diseases for the chicken by itself (Salmonella enterica subsp. Gallinarum), and this pathogen under particular conditions as paratyphoid Salmonella could become highly relevant for the health and welfare of human been. Indeed, it has become one of the most important items under alertness; Salmonellosis is a top disease in a new concept, "One World, One Health". It is based on the understanding that humans, animals, and the environment are inextricably linked. The health of humans relies on well-functioning ecosystems, which provide, among others, food security as well as limit disease. Salmonellosis is regarded as a foodborne infection of the gastrointestinal tract of the animal and human intestines. This pathogen is frequently spread to animals and humans through contaminated animal or human feces or food. Nowadays, eggs and poultry meat as the main human food worldwide show a high potential for foodborne infections. As a matter of fact, a large proportion of humans become infected most frequently through contaminated poultry food, and elderly and children infected with paratyphoid Salmonella could be under a high death risk. The authors reviewed here the most important factors related to the ecosystem at the chicken gut level that tried to explain why Salmonella continues to be the most important pathogen involved in this unfortunate but very common foodborne poultry disease. This latter issue is the most strength of the current review. Otherwise, although the current MD covers almost all crucial knowledge about the original author's proposal, the weakness of this work is that it does not have a final analysis or remarks and does not display any conclusion.

Scientist approach concept: The current manuscript draft (MD) is well written and addresses important items about the persistence of a critical foodborne illness like Salmonellosis from Poultry.

Specific approach comments: This MD has some observations.

Title:

"Interactions between the intestinal ecosystem and Salmonella persistence in the poultry intestine."

The authors ought to be more specific with their title; for example, what kind of interactions do the authors want to address here? Physiological, immunological, pathological, symbiotic?

I think they need to change the next expression in their title, intestinal ecosystem, to another more accurate expression, e.g., gastrointestinal ecosystem. It also could apply to the poultry intestine, and another expression could change it.

The author should define what kind of Salmonella they are discussing here: Avian Typhoid, Paratyphoid Salmonella? These two types of Salmonella affect Poultry birds.

The authors should choose the appropriate words when writing a title based only on a general description of the interactive relationship between poultry chicken microbiota at the intestine and the persistence of Salmonella sp in the Poultry. The word, interactions, drives here to a very overall and undefined action, and as a result, it needs to be a clear title.

The main interactions that authors describe here are between the immune host and the features of the pathogen and do not with the resulting fact of Salmonella persistence. The authors must review their title again carefully to be sure that they accurately describe the scope of review they are trying to communicate to the potential readership. Authors must choose another title that better addresses their main review goals. Authors should modify their title, which has to be more appropriate to their MD content.

Abstract:

Line 8. poultry has to be complete to: poultry birds.

Line 11. Please verify all spaces after the point between sentences throughout your MD.

Line 15 and 16. The initial phase (4-48 hours) is until 2 days, so how do the authors define the period between the initial and second phase when the latter outset from 4-14 days? They have lost two days here. This should be clarified.

Line 17. Expansion or increase?

Line 17 to 20. What about macrophage's role in Salmonella infection or translocation? Is not important enough here?

Line 21. unnecessary word here: in Poultry

Line 23. disease tolerance or Salmonella infection? This sentence should paraphrased in order to be the most comprehensive.

Line 23. Results about what? Immunometabolic reprogramming or gastrointestinal ecosystem interactions?

Keywords:

Searchable online databases often include all the words in the title in search queries. This means that if someone types a word that is found in your title into a search field, your paper will be listed in the search results only with this word; if you repeat it in your Keywords, you are limiting search queries, so you do not have to repeat words from the title in your Keywords So, in the Keywords section, authors should not repeat words from the title. The authors should review and suggest new Keywords. Please do not repeat words like Salmonella or persistence.

1.0 Introduction:

Line 30- 31. References?

Line 36. What is the next superscript: short-lived7?

Line 36. Reference?

Line 37. concentrations?

Line 42 and 46. overcoming is repeated in the same paragraph, find another synonym that means this same important feature of Salmonella in chickens appropriately.

Line 48. Please review the syntax for fluency in this sentence.

1.1. Gut Health

57. Glycocalyx creating?

Line 58-59. to perform is repeated in the same sentence; find another synonym to address the same intent of this sentence.

Line 63. regulate? Please review the intended meaning of this paragraph.

1.2. The gut ecosystem

Line 69. epithelial cells desquamation or just as lining cells?

Line 71. exposing or displaying?

1.3. Components of the intestinal ecosystem

Line 78-80. How about the nervous control of the physiological gut process?

Line 85- 86. Check this space between lines.

Line 87-88. Claudin 1- 2, occludin? Please analyses it in a deeper way

Line 91. Define PRRs before use later in the writing

Line 98- 99. Check this space between lines.

Line 102. Memory B-cells are not professional antigen-presenting cells at lamina propria.

Line 100-105. Need a reference for Poultry.

Line 105-108. Need a reference for all these immunological features in Poultry.

Line 116. Local immune defence? It needs a valid reference at the end of this sentence.

Line 115. Verify the space after the point

Line 121. Add a point

Line 125-126. Information stated before in lines 115-116 is repeated; verify it.

Line 128. Why is this expression underlined?

Line 130. Verify: endocrine system and the CNS to the brain

Line 134. Please review the syntax for fluency in this sentence.

Line 148. Verify the spaces after the point throughout your MD

Line 150-151. ingested food is not from the host, verify this statement

Line 152. carbohydrates or sugars?

Line 153. potential beneficial? potential beneficial microorganism, maybe? Clarify, please

Line 159-162. Refrain from mentioning anything about the competitive exclusion principle here. The Nurmi concept is not important in this paragraph? Has all the above already been overcome? If so, please indicate why; if not, is there no recent research about it?

Line 169-173. Caballero-Flores et al., 2022 stated about competitive microbial–microbial interactions and induction of host immune responses; in this paragraph, you stated about the former but nothing about the latter.

Line 174-176. References?

Line 177-179. OK, but how about the autochthonous microbial metabolite's role in the chicken intestinal pH? Moreover, how to do novel dietary-protected organic acids function in the gut ecosystem context?

Line 180-182, Would you like to talk something about mechanisms to promote epithelial cell barrier function from the dietary tryptophan? or inhibition of pro-inflammatory cytokine production by the breakdown of dietary arginine?

2. Salmonella Interactions with the Intestinal Ecosystem in Chickens

Line 184. burden or losses?

Line 186. How about Salmonellosis worldwide?

Line 189. Is writing the species with the first letter in capital letters correct? Is it a new rule from Bergey's manual? Please verify it

Line 189-191. This statement is not accurate enough because some phagotypes of this two parathyroid Salmonella can cause clinical signs in chicks under three weeks of age, even mortality (e.g. Se PT 4 (ch. CA) with 23.3% of mortality, reference: Dillon et al., Avian Diseases 43: 506-515, 1999).

Line 197.

Research indicates that inflammation is built through an inflammasome, a large multimolecular complex. Please specify what part of this complex you are referring to here.

Line 199. Table 1 is unclear; please improve it to become more friendly for potential readers of your paper. It has not subtitle for every column. A detailed description of each subtitle into every column of the table is necessary.

2.1. Salmonella and Host Inherent Factors

Line 205. How about PAMPs role at the outset of Salmonella infection? How about the MyD88-dependent TLR signalling in St? You can review other interesting items like those; it could enrich your work.

Line 208-212. This statement is crucial to justify your current review, but do you have references?

Line 213. Disease resistance or colonization resistance? Disease resistance through colonization resistance?

Line 213-214. Review syntax for fluency in this sentence.

Line 216. It needs to be clarified that you're quoting (1) here, as well as the number (2) in the next line. Correct it, or review syntax for fluency. It is better if you use: (i), (ii), (iii), (iv) and (v).

Line 226 expansion or increase?

Line 232. Why underline this kind of bird?

Line 234. What phase 2-4 days PI is? First one, second, or is it the transition between them?

Line 235. Why underline AMPK and not mTOR?

Line 237-238. Why underline this biochemical step?

Line 240. Is there any other reference different to these self-citations?

Line 240. Why underline that biochemical expression?

Line 241, You stated here that the third phase appears to begin shortly after d 4 post-infection, but above in line 225 you wrote that the second phase is from 4 to 14 days post-infection, it is unclear. Please verify the appropriate order for Salmonella post-infection phases.

Line 242. Why underline homeostasis?

Line 242. In the substantive Treg s reg is aligned slightly below the baseline in subscript format? Is that correct? Because you above in line 227 wrote it as Tregs, in normal format.

Line 241-244. References?

Line 247. Why underline microbiome?

Line 246-248. There are some reptilians like Asiatic turtles or snakes where some species of Salmonella could be considered as a part of "normal" microbiome. Can you discuss something like that under your own hypothesis or speculation? it could enrich your review.

2.1.1 Enteric neuroendocrine system.

Line 249. This subtitle needs to be clearer in this part of your MD because in Line 109, you already talked about it; if you need to use this in the current context, please modify it accordingly.

Line 252. You defined Tregs above, and here you define it again; it is unclear.

Line 258. Present tense? Demonstrates or demonstrated? Please check the appropriate style to write a review paper.

Line 260. Why underline these two words?

Line 260-275. Check the line spacing of this paragraph; it has another format

Line 270. Is the ERK pathway an article without capital letter?

Line 273-274. It is commonly shown in mice; do you have a suitable reference for this immunological action in chickens?

Line 277-279. It is not clear because some phagotypes of the parathyroid Salmonella elicit clinical signs in chicks; even parathyroid Salmonella like Salmonella enterica serovar Enteritidis can produce mortality (e.g. Se PT 4 (ch. CA) with 23.3% of mortality, reference: Dillon et al., Avian Diseases 43: 506-515, 1999).

Line 283-285. Reference for inhibition of anorexia but in Poultry

2.2. Microbiota-linked factors

Line 287. There is no stomach in birds because proventriculus and gizzard cover the function of a unique compartment that in mammals with just one stomach (monogastric) this is composed by only one compartment like in pigs, cats or monkeys.

Line 288. Is American or England your style of writing? If it is American, please uniform it throughout the MD. Is avian ceca does not avian caeca. There is no colon in birds.

Line 289. Check the proper closing of your parentheses

Line 299. What are the differences between genetics and genetic lines?

Line 325-327. This reference is old; you should find a most suitable and recent research for this statement. The genus or individual species of the bacteria are not used more as the overall explanation for microbiota dynamic through the life of birds; currently, with the help of Next-generation sequencing (NGS), this kind of report are based on Taxa families, mainly

327-330. Taxa, families. This is a current reference. It is OK

331-334. This is a current reference. OK

337, Salmonella enterica serovar Enteritidis? Verify current nomenclature by referring to the Bergey's manual

Line 367. Is Shannon index (α-diversity) or Line 370. Chao1 index (α-diversity)? It is not clear enough.

Line 371. You wrote here: one-day-old, but above in line 356 you wrote 1-day-old, and in line 320 you wrote day-of-hatch. Try to uniform this expression through the MD.

Line 398. Are you referring to: Salmonella enterica serovar Typhimurium? Verify if the current nomenclature you are using here in your MD is adequate. You could refer to the Bergey's manual, because the scientific name format of these microorganism change more frequently than we expect.

Line 404-407. This paragraph is not clear enough: "Salmonella Typhimurium challenge reduced the significant increase of Lactobacillus observed in the ileum of non-challenged animals." Lastly, chicks where the Lactobacillus was reduced, were they Salmonella enterica serovar Typhimurium challenged or non-challenged?

Line 407-409. You need more than just one reference for this statement.

Line 402. Specifically, what immune response dynamic changes are referring to?

Lines 413-415. How about Clostridia? This population increases or decreases when St colonizes the chicken ceca at this time?

Line 421. Please review: Erysipelatoclostridium. I am no sure if this type of genus exists or is common to report together both of them, or well, it belongs to a TAXA family; in this case, you must specify this one like this, do not as genus. I know that the genus Erysipela is just one, and the genus Clostridium is another one different from the former.

Line 421-427. This paragraph is redundant with the paragraph in lines 413-415 and 418-421; please try to rewrite it in just one paragraph.

Line 433-434. Here, you are speculating. It is not bad, but you need a reference supporting this speculation.

Line 435. The gerund and past-participle form of the verb: consider is repeated here; consider re-paraphrasing this sentence.

Line 445-447. Can the pH function as a "habitat filter"? If yes, how does it act at this level?

Line 447-449- You need a reference for this statement.

Line 450-458. The authors need to write a brief conclusion for this paragraph, it needs to be completed.

Line 467-470 . What type of microorganisms have this downregulation event, potential beneficial or pathogens?

Line 470-472 . What type of microorganisms shows this perturbation, beneficial or pathogens? Mention which they are and the degree of affectation owing to the Se infection.

Line 474. How is the lactate high or low?

Line 472-477. You mentioned before that the high presence of this bacteria (Enterococcus) could be used as a fingerprint to exclude any Salmonella infection in chicks. How is connect that statement with the current information from this paragraph?

Line 487. α-diversity or β-diversity?

488-490. Why? Explain briefly, because the paragraph from Lines 490-494 contradicts the idea that the restriction of BCAA resulted in increased susceptibility to St. Is it not a consequence instead of a cause?

Line 499-500. It is a conclusion of the above paragraphs, not a separate paragraph.

Line 511. What is TCA?

Line 516. There is no conclusion for the paragraph from lines 459-516.

Please verify the format for references according to Animals Journal (MDPI), because some were inappropriately written. E.g. lines 531, 674.

FINAL REMARK: This work has not final remarks and conclusions. The authors must approach an appropriate closing of their review under an attractive style of writing that could be adressed through a smart analysis of all this valuable information.

Reviewer 2 Report

Please carefully review the article again.  There are many sentences that need to be restructured, are missing words, or need grammatical editing.  Additionally, the content could be improved and restructured which is why comments end at Table 1.  Table 1 includes information that is not mentioned anywhere else in the article.  There is also no summary or conclusion paragraph to tie everything together.  The article contains meaningful information but has many sections that are not clear/concise.

Line 11: Understanding of how the pathogen interacts – insert how (this is an example of where a word has been omitted and it needs to be fixed)

Line 12: Suggest removing ‘robust’

Lines 16: Initial phase of what? Immune response I assume. Please correct.

Lines 23-25: What results? Please correct.

Line 25: Suggest modifying to ‘asymptomatic carriage of Salmonella’

Line 33-40: This needs to be broken into 2-3 sentences for clarity. There are multiple sections throughout the article with this similar problem.

Line 48: “and it is these” – again a word is missing.  I will not address those moving forward.  Please just read carefully and fix.  Thank you!

Lines 61-63: Suggest removing this sentence or restructuring.  At present, it does not add any meaningful information to the article.

Line 66-67: Please add a few more environmental factors.

Lines 51-72: Gut health and gut ecosystem sections could be combined and remove lines 70-72.

Recommend using “host factors” instead of “host inherent factors” since this is understood.

Formatting of section 1.3 could be adjusted for clarity for each component/factor of the intestinal ecosystem.

Spacing issues Line 85-86, 98-99. I will not mention again. Please just carefully review the article.

Line 136: Remove ‘gastrointestinal’ and use GIT consistently throughout the manuscript.

Why include lines 78-182 if you referred to another review article?  This section could be truncated or simply summarized since you referred to another article.

Line 185: 48 million people? Line 184 and 185 are contradictory.  Please provide more context.

Table 1. Difficult to follow. Please include descriptive headings, references, etc. What is the purpose of the table? There is a lot of information in the table that is nowhere to found in the text…

See comments listed above.

Round 2

Reviewer 1 Report

At this time, my advice is that the current manuscript draft animals-2550598-peer-review cannot published like it looks now, this MD shows some minor faults but all of them can be solved giving an appropriate layout for final version.

- e.g. some connector are italicized like in page 8/16 " siella, Oscillospira, Holdemania and Coprococcus, and increased Klebsiella and Escherichia".

- In page 5 of the new version of MD, the last paragraph (B) Salmonella and neurotrophin signaling (11 lines)… has a serious problem of format.

-The figure 1 include in this new version of MD did not mention anywhere in the text, and it has not a clear explanation.

-Some words like AMPK kept underline and another do not (e.g. mTOR), this underline of words was done with not clear reason, and  persist throughout MD.

The MD need another final syntax review in order to avoid any mistake or fluency comprehension

Authors should correct all minor faults before their paper could be consider to publish in animals.

Reviewer 2 Report

I have reviewed the author's revised manuscript and approve.

I have reviewed the author's revised manuscript and approve.